# Influence of Time-Lag Effects between Winter-Wheat Canopy Temperature and Atmospheric Temperature on the Accuracy of CWSI Inversion of Photosynthetic Parameters

**DOI:** 10.3390/plants13121702

**Published:** 2024-06-19

**Authors:** Yujin Wang, Yule Lu, Ning Yang, Jiankun Wang, Zugui Huang, Junying Chen, Zhitao Zhang

**Affiliations:** 1College of Water Resources and Architectural Engineering, Northwest A&F University, Xianyang 712100, China; wangyujin@nwafu.edu.cn (Y.W.); luyule@nwafu.edu.cn (Y.L.); yangning1228@126.com (N.Y.); wangjiankun@nwafu.edu.cn (J.W.); hzgui@nwafu.edu.cn (Z.H.); junyingchen@nwafu.edu.cn (J.C.); 2Key Laboratory of Agricultural Soil and Water Engineering in Arid and Semiarid Areas, Ministry of Education, Northwest A&F University, Xianyang 712100, China

**Keywords:** time-lag effects, winter wheat, CWSI, photosynthetic rate, transpiration rate, stomatal conductance

## Abstract

When calculating the CWSI, previous researchers usually used canopy temperature and atmospheric temperature at the same time. However, it takes some time for the canopy temperature (Tc) to respond to atmospheric temperature (Ta), suggesting the time-lag effects between Ta and Tc. In order to investigate time-lag effects between Ta and Tc on the accuracy of the CWSI inversion of photosynthetic parameters in winter wheat, we conducted an experiment. In this study, four moisture treatments were set up: T1 (95% of field water holding capacity), T2 (80% of field water holding capacity), T3 (65% of field water holding capacity), and T4 (50% of field water holding capacity). We quantified the time-lag parameter in winter wheat using time-lag peak-seeking, time-lag cross-correlation, time-lag mutual information, and gray time-lag correlation analysis. Based on the time-lag parameter, we modified the CWSI theoretical and empirical models and assessed the impact of time-lag effects on the accuracy of the CWSI inversion of photosynthesis parameters. Finally, we applied several machine learning algorithms to predict the daily variation in the CWSI after time-lag correction. The results show that: (1) The time-lag parameter calculated using time-lag peak-seeking, time-lag cross-correlation, time-lag mutual information, and gray time-lag correlation analysis are 44–70, 32–44, 42–58, and 76–97 min, respectively. (2) The CWSI empirical model corrected by the time-lag mutual information method has the highest correlation with photosynthetic parameters. (3) GA-SVM has the highest prediction accuracy for the CWSI empirical model corrected by the time-lag mutual information method. Considering time lag effects between Ta and Tc effectively enhanced the correlation between CWSI and photosynthetic parameters, which can provide theoretical support for thermal infrared remote sensing to diagnose crop water stress conditions.

## 1. Introduction

The timely and accurate diagnosis of crop water stress conditions effectively determines the timing of irrigation and facilitates precision irrigation, crucial for enhancing water use efficiency (WUE) and increasing yield [1,2]. When crops experience water stress, their physiological indicators, such as photosynthetic parameters, leaf water potential, and external morphology, undergo changes. These changes include a decrease in the leaf area index, a reduction in chlorophyll concentration, and a diminution in leaf length and width [3,4]. Physiological indicators, including photosynthetic parameters, leaf water potential, and stem water potential, focus on the crop itself for research and offer a straightforward, scientific method to diagnose the status of crop water deficit. These indicators have proven effective in monitoring the water status of crops [5]. Moisture stress impacts photosynthetic parameters across the reproductive period of crops, displaying consistent trends in the net photosynthesis rate (Pn), transpiration rate (Tr), and stomatal conductance (gs), which all decrease with increasing moisture stress [6]. The extent of changes in photosynthetic parameters varies with different levels of water stress, with minor reductions under mild water stress and significant declines under moderate to severe water stress [2].

Beyond the physiological indicators of crop water deficit, the crop water stress index (CWSI), sensitive to soil moisture, stands as a reliable indirect metric for monitoring crop water status. Under water stress conditions, crops exhibit reduced stomatal conductance and diminished transpiration cooling, leading to an increase in canopy temperature. Idso et al. [7] found that the differential in canopy temperature following noon effectively measures the crop water deficit, revealing a consistent linear relationship between the canopy air temperature differential (CTD) and vapor pressure deficit (VPD) under clear sky conditions, which is not affected by environmental factors, such as wind speed. In response to the above phenomenon, Idso, Jackson, Pinter, Reginato and Hatfield [7] propose the CWSI empirical model, which has the advantages of fewer computational parameters, ease of measurement, and sensitivity to crop varieties. Jackson et al. [8] proposed the CWSI theoretical model based on the principle of canopy energy balance, taking into account environmental factors such as aerodynamic resistance, crop minimum canopy resistance, and net radiation, making the CWSI model more theoretical.

The CWSI is a sensitive indicator used to reflect water stress caused by the stomatal function of the crop, and continuous water stress results in an increasing trend of the CWSI and a decreasing trend of Pn, Tr, and gs [9]. There is a good negative correlation between the CWSI and photosynthetic parameters [6,10]. The results of Ramos-Fernández et al. [11] showed a strong correlation between the CWSI and gs (R^2^ = 0.91). When the crop is subjected to water stress, the soil–root hydraulic resistance increases [12], which reduces root water transport and eventually leads to the reduction in or closure of plant stomata and a decrease in photosynthetic parameters [13]. Different physiological characteristics of wheat have different sensitivities to soil moisture [14]; therefore, the correlation between Pn, gs, Tr, and CWSI varies.

In calculating the CWSI, previous researchers always used atmospheric temperature (Ta) and canopy temperature (Tc) at the same moment [15]. However, there is a time-lag effect in the response of Tc to Ta [16]. Therefore, it is more accurate to use atmospheric temperature that actually influences the canopy temperature. Zhang et al. [17] discovered that accounting for the time-lag effect significantly enhances the accuracy of the CWSI in estimating soil water content. Currently, research on the impact of this time lag on the accuracy of the CWSI in the inversion of photosynthetic parameters remains unexplored.

We hypothesized that the time-lag effects between the canopy temperature and atmospheric temperature have a significant impact on the model accuracy of the CWSI inversion of photosynthetic parameters. Therefore, we conducted an experiment with winter wheat, where we continuously monitored the canopy temperature and environmental factors of winter wheat. We quantified the time-lag parameters between Ta and Tc using time-lag peak-finding, time-lag cross-correlation, time-lag mutual information, and time-lag gray correlation analysis. We then modified the theoretical and empirical CWSI models based on these time-lag parameters. Finally, we investigated the implication and mechanisms of Ta and Tc time-lag effects on the accuracy of the CWSI inversion of photosynthesis parameters.

## 2. Results

### 2.1. Time-Lag Parameters of Winter Wheat under Different Water Stresses

As depicted in Figure 1, the CCE equation fitted the daily variation process of winter-wheat canopy temperature smoothed by S-G filtering with excellent accuracy (R^2^ = 0.98), and the ECS equation fitted the daily variation process of atmospheric temperature smoothed by S-G filtering with equal precision (R^2^ = 0.98). As seen in Figure 2, Figure 3, Figure 4, Figure 5 and Figure 6, among the time-lagged parameters of Tc and Ta obtained by different methods, the gray time-lag correlation analysis was the largest. The time-lag peak-seeking method and the time-lag mutual information method were the second largest, followed by the time-lag cross-correlation method.

The time-lag parameters between the canopy temperature (Tc) and atmospheric temperature (Ta), calculated using four different methods, exhibited distinct values across varying irrigation treatments. For the fully irrigated treatment, the time-lag parameters were approximately 53 min, 44 min, 58 min, and 97 min when calculated using the time-lag peak-finding method, time-lag cross-correlation method, time-lag mutual information method, and gray time-lag correlation analysis, respectively; for the mild water stress treatment, these time-lag parameters were about 52 min, 43 min, 55 min, and 92 min, respectively. For the moderate water stress treatment, the parameters were approximately 55 min, 44 min, 54 min, and 98 min. Lastly, for the severe water stress treatment, the parameters were around 44 min, 32 min, 42 min, and 76 min. These results highlight the variability in time-lag parameters across different irrigation treatments, as well as the influence of the chosen calculation method.

This indicates that the time lag between the Tc and Ta obtained from different calculation methods for the fully irrigated, mild water stress, and moderate water stress treatments did not differ significantly. However, for the severe water stress treatment, the Tc reached its peak time later, resulting in a decrease in the time-lag parameter between the Ta and Tc by approximately 10 to 22 min. This phenomenon might be associated with the soil moisture threshold [18]. When the soil moisture threshold was reached, the water lost through transpiration in winter wheat could not be replenished promptly. To ensure the normal life activities of the crop, the expansion rate of the crop leaves was reduced, stomatal conductance decreased significantly, transpiration rate declined, and canopy temperature continued to increase, reaching the peak time later. This led to a shorter time lag between the atmospheric temperature and canopy temperature [19].

In addition, the cross-correlation coefficient, mutual information coefficient, and gray correlation coefficient values corresponding to the peak moments for the fully irrigated, mild water stress, moderate water stress, and severe water stress treatments did not differ significantly. This indicated that the linear correlation [20], nonlinear correlation [21], and curve similarity [22] of Tc and Ta under the four moisture treatments after a time-lag correction did not differ much.

The accuracy of the CWSI inversion of photosynthetic parameters before and after time-lag effects was considered.

### 2.2. Time-Lag Peak-Seeking Method, Time-Lag Cross-Correlation Method, Time-Lag Mutual Information Method, and Gray Time-Lag Correlation Analysis

As shown in Figure 7 and Figure 8, correcting the time lag between the Ta and Tc improved the accuracy of the CWSI inversion for Pn. After CWSI empirical and theoretical models were corrected using the time-lag peak-seeking method, time-lag cross-correlation method, time-lag mutual information method, and gray time-lag correlation analysis, the correlation between the CWSI and Pn improved for all methods, with the empirical model showing a more significant improvement. This indicated that the time-lag effect had a substantial impact on the accuracy of the CWSI empirical model in inverting Pn, while its impact on the accuracy of the CWSI theoretical model in inverting Pn was small and negligible.

As shown in Figure 9 and Figure 10, the time-lag effect has a small impact on the accuracy of the CWSI theoretical model inverting Tr and a large impact on the accuracy of the CWSI empirical model inverting Tr. The correlation between the CWSI and Tr does not change after correcting the CWSI theoretical model by using the time-lag peak-seeking method. The correlation between the CWSI and Tr is improved by correcting the CWSI theoretical model by using time-lag cross-correlation method, time-lag mutual information method, and gray time-lag correlation analysis. The correlation between Tr and the CWSI theoretical model corrected based on gray time-lag correlation analysis and time-lag mutual information is the best (R^2^ = 0.90). The correlation between the CWSI empirical model and Tr decreases after correcting the CWSI empirical model using the time-lag mutual information method and gray time-lag correlation analysis. The accuracy of the CWSI inversion of Tr improves after correcting the CWSI empirical model using the time-lag peak-seeking method and time-lag mutual correlation method. The correlation between Tr and the CWSI empirical model corrected based on the time-lag mutual correlation method is the highest (R^2^ = 0.94).

As demonstrated in Figure 11 and Figure 12, the correlation between the CWSI and gs remains unchanged after the CWSI theoretical model was corrected by applying the time-lag peak-seeking method. However, the correlation between the CWSI and gs improved after the CWSI theoretical model was corrected using the time-lag cross-correlation method, the time-lag mutual information method, and the gray time-lag correlation analysis. Notably, the time-lag mutual information method enhanced the accuracy of the CWSI theoretical model inversion of gs the most (R^2^ = 0.96). The time-lag effect significantly impacted the accuracy of the CWSI empirical model inversion of gs. The correlation between the CWSI and gs increased after correcting the CWSI empirical model using the time-lag peak-seeking method, the time-lag mutual correlation method, the time-lag mutual information method, and the gray time-lag correlation analysis. gs showed the highest correlation with the CWSI empirical model based on the time-lag mutual information method (R^2^ = 0.96).

In summary, it was observed that time-lag effect between the Ta and Tc caused a significant impact on the accuracy of the CWSI inversion of photosynthetic parameters. The impact was more substantial on the accuracy of the CWSI empirical model for inverting photosynthetic parameters, and the time-lag-corrected CWSI empirical model demonstrated a higher correlation with the photosynthetic parameters. This indicated that the CWSI empirical model was more sensitive to the time-lag effect than the CWSI theoretical model. The reason for this phenomenon might be that the CWSI theoretical model required measurements of net radiation, soil heat flux, wind speed, and canopy resistance, making the theoretical models less volatile [23]. The correlation of the time-lag-corrected CWSI with Pn, Tr, and gs was in the order of gs > Tr > Pn. Pn showed the highest correlation with the empirical/theoretical CWSI models corrected by the time-lag mutual information method (R^2^ = 0.8); Tr had the best correlation with the CWSI empirical model corrected by the time-lag mutual information method (R^2^ = 0.93); and gs exhibited the best correlation with the CWSI empirical model corrected by the time-lag mutual information method (R^2^ = 0.93). Occasionally, time-lag correction reduced the accuracy of the CWSI inversion of photosynthetic parameters. This reduction could be attributed to the fact that the time-lag effect between the Tc and environmental factors, such as relative humidity and solar radiation, was not accounted for [24].

Meanwhile, the time-lag effect was the result of the continuous direct or indirect influence of previous environmental factors on crops, representing an accumulative process [25,26,27]. The time-lag peak-finding method, which utilized a function to fit the daily change curves of the Tc and Ta and defined the time-lag parameter solely by the time difference between its peak points, exhibited certain limitations.

### 2.3. Machine Learning Algorithms for Predicting CWSI Empirical Models Based on Time-Lag Mutual Information Correction

The accuracy of the CWSI empirical model corrected based on the time-lag mutual information method for the inversion of photosynthetic parameters was overall high. It was investigated by using Genetic Algorithm Optimized Support Vector Machines (GA-SVMs) based on genetic algorithms, Bayesian Optimized Long and Short-Term Memory Neural Networks (Bayes-LSTMs), Particle Swarm Algorithm Optimized Long and Short-Term Memory (PSO-LSTM) based on particle swarm algorithms, Convolutional Bi-directional Long and Short-Term Memory Neural Networks (CNN-BILSTMs), Attention Mechanism Long Short-Term Memory Neural Networks (attention-LSTMs), and Attention Mechanism Gated Recurrent Unit (attention-GRU) machine learning algorithms for predictions. The prediction accuracies are shown in Table 1.

With GA-SVM > Bayes-LSTM > PSO-LSTM > CNN-BILSTM > attention-LSTM > GRU-attention. The prediction accuracy of the above models for the CWSI empirical model corrected by the time-lag mutual information method was higher overall. Predicted effect diagrams are shown in Figure 13, Figure 14, Figure 15, Figure 16, Figure 17, Figure 18, Figure 19, Figure 20, Figure 21, Figure 22, Figure 23 and Figure 24. The GA-SVM model had the highest prediction accuracy (R^2^ = 0.982, RMSE = 0.017).

## 3. Materials and Methods

### 3.1. Study Site Description

The experimental site is located at the Institute of Water Saving Agriculture in Arid Regions, Northwest Agriculture and Forestry University, Yangling District, Shaanxi Province, China (108°24′ E, 34°20′ N). This region is characterized by a warm, temperate, semi-humid monsoon climate, distinguished by four distinct seasons and moderate rainfall. The average annual temperature ranges from approximately 13 °C to 15 °C. Rainfall predominantly occurs in July and August, driven by the southeast monsoon, with an annual average between 600 mm and 800 mm. The effect of groundwater recharge is not considered in this experiment.

### 3.2. Experiment Design

The experimental site was 32.5 m × 10.5 m and divided into 12 plots, each measuring 4 m × 4 m. Protected row treatments were used to mitigate the effects of water infiltration (Figure 25). Four moisture treatments were used in the experiment to obtain generalizable results: T1 (fully irrigated), T2 (mild water stress), T3 (moderate water stress), and T4 (severe water stress). The upper irrigation limits were set at 95%, 80%, 65%, and 50% of the field water holding capacity for T1, T2, T3, and T4, respectively. Each moisture treatment was replicated three times. Instruments for the continuous monitoring of canopy temperature and environmental factors were positioned above experimental plots 2, 5, 8, and 11. The cultivar used was “Genmai 68” winter wheat sown at 25 cm spacing with 30 g of seed per row. The sowing date was 19 October 2022 and the harvest date was 1 June 2023. Irrigation was carried out by drip irrigation system, and the irrigation quota is detailed in Table 2. The measured volumetric soil water content was calculated by oven-drying method and the irrigation quota is calculated as follows:(1)$$m=H\cdot (\theta_{s}-\theta_{o})\cdot p\cdot s$$
where *m* is the irrigation quota (mm); *H* is the planned wetted layer depth (m): 0.4–0.5 m (green-up stage), 0.5–0.6 m (jointing stage), 0.6–0.8 m (tasseling stage), and 0.8–1.0 m (grouting period); *θ_s_* is the field capacity (%), which is the upper limit of soil moisture content; *θ_o_* is the measured volumetric soil moisture content (%); *s* is the trial plot size (m); and *p* is the drip irrigation wetting ratio, 0.6. 

### 3.3. Data Acquisition

#### 3.3.1. Tc Measurements

In this study, the canopy temperature of winter wheat was continuously monitored using an SI-411 infrared thermometer. The monitoring interval was set at 2 min. Considering the effect of crop cover on the instrumental monitoring of canopy temperature, the time-lag parameter was calculated in this experiment starting from 16 February 2023. The canopy temperature of winter wheat for the four moisture treatments is shown in Figure 26.

#### 3.3.2. Environmental

In this experiment, meteorological factors were continuously monitored using the AWS-CR1000 scientific-grade automatic meteorological monitoring system, as detailed in Table 3. Data collection intervals were set at 2 min. Meteorological factors are shown in Figure 27.

#### 3.3.3. Photosynthetic Parameters Measurements

The differences in crop physiological indicators at different irrigation levels were small in the morning and evening, and the differences were largest around midday, which could accurately reflect the crop water status [28,29]. Therefore, we chose sunny and windless days to collect the photosynthetic parameters of winter wheat: stomatal conductance gs (mol/(m^2^·s)), net photosynthesis rate Pn (µmol/(m^2^·s)), and transpiration rate Tr (mmol/(m^2^·s)) at 14:00 using a portable photosynthesizer model Li-6800 from LICOR, Lincoln, NE, USA. Three wheat plants were randomly selected from each plot, and the measurements were repeated three times for each wheat flag leaf, and the average value was taken as the photosynthetic parameters of the crop under the moisture treatment; to ensure the accuracy of the acquired data, the CO_2_ concentration of the Li-6800 portable photosynthesizer reached 400 μmol/mol, and the intensity of the light reached 1000 μmol/(m^2^·s) during the measurement. The data of photosynthetic parameters were collected 12 times in this experiment (Figure 28).

### 3.4. Data Processing

#### 3.4.1. Savitzky–Golay (S-G) Filter

The Savitzky–Golay (S-G) filter [30] is a smoothing filtering technique that employs local least squares to eliminate noise from time-series data. This method achieves its smoothing effect by fitting a polynomial to the data, which effectively removes noise while preserving the signal’s original shape as closely as possible. Consequently, the S-G filter maintains the integrity of the signal, ensuring effective smoothing.

#### 3.4.2. Z-Score Normalization

Employing the Z-Score standardization method [31], the dimensionless standardization of raw indicator data effectively mitigates the impact of discrepancies in data size, characteristics, and distribution. This approach eliminates unit differences across the data, enabling comparability among variables with diverse characteristics, while preserving the original distribution pattern of the data.

#### 3.4.3. Time-Lag Peak-Seeking Method

The time-lag peak-seeking method [32,33] selects the appropriate function to fit the Ta and Tc and determine the peak position of the fitted curve. The time difference corresponding to this peak point represents the time-lag parameter between the Ta and Tc. Zhang and Wu [34] used the Gaussian function to fit the canopy temperature and atmospheric temperature of summer maize and achieved good accuracy. However, the Gaussian function fits the canopy temperature and atmospheric temperature of winter wheat with lower accuracy. The CCE equation has a higher fitting accuracy for the canopy temperature after smoothing by S-G filtering, and the ECS equation has a higher fitting accuracy for the atmospheric temperature after smoothing by S-G filtering.

The CCE equation is expressed as follows:(2)$$double1\;z=x-{xc}_{1}$$
(3)$$\begin{array}{l} y=y_{0}+A\times (\exp(-z\times z/(2\times w))+(1-0.5\times (1-\tan(k_{2}\times (x-xc_{2}))))\times \\ B\times \exp(-0.5\times k_{3}\times (abs(x-xc_{3})+(x-xc_{3})))) \end{array}$$
where *xc*_1_ is the peak moment of winter-wheat canopy temperature. The fitting accuracy was judged by the coefficient of determination (R^2^).

The ECS equation is expressed as follows:(4)$$\begin{array}{l} y=y_{0}+A/(w\times sqrt(2\times pi))\times (exp(-0.5\times {\left((x-xc)/w\right)}^{2})\times (1+(a_{3}/(3\times 2\times 1))\times ((x-xc)/w)\times ({\left((x-xc)/w\right)}^{2}-3)+\\ (a_{4}/(4\times 3\times 2\times 1))\times ({\left((x-xc)/w\right)}^{4}-6\times {\left((x-xc)/w\right)}^{3}+3)+((10\times a_{3}^{2})/(6\times 5\times 4\times 3\times 2\times 1))\times ({\left((x-xc)/w\right)}^{6}-\\ 15\times {\left((x-xc)/w\right)}^{4}+45\times {\left((x-xc)/w\right)}^{2}-15))) \end{array}$$
where *xc* is the moment of peak atmospheric temperature. The accuracy of the fit is judged by the coefficient of determination (R^2^), and the peak time difference between Tc and Ta is the time-lag parameter between Tc and Ta.

#### 3.4.4. Time-Lag Cross-Correlation Method

Zhang et al. [35] used the time-lag cross-correlation method to calculate the time lag between the canopy temperature and atmospheric temperature in winter wheat. They then found that correcting the time-lag effect between Tc and Ta by the time-lag cross-correlation method can improve the accuracy of the CWSI inversion of SWC. X (Tc) is first mapped to Y (Ta) in the chronological order of observations. Then, Tc is shifted in steps of 2 min and the Pearson correlation coefficients of the two series are calculated. When the Pearson correlation coefficient attains its maximum value, the corresponding shift duration is designated as the time-lag parameter for the two series [16,36], where the correlation coefficient is calculated as:(5)$$R_{k}=\frac{\sum_{i=1}^{n-k}\left(x_{i}-\bar{x_{i}}\right)(y_{i+k}-\bar{y_{i+k}})}{\sqrt{\sum_{i=1}^{n-k}{\left(x_{i}-\bar{x_{i}}\right)}^{2}}\sqrt{\sum_{i=1}^{n-k}{\left(y_{i+k}-\bar{y_{i+k}}\right)}^{2}}}$$
(6)$$R_{m}=\max(R_{k})$$
(7)$$T_{L}=2m$$
where *R_k_* is Pearson correlation coefficient for a sliding shift number of *K*; *n* is the sample size; $x_{i}$ is the canopy temperature (°C); $y_{i+k}$ is the atmospheric temperature (°C); $\bar{x_{i}}$ is the mean of canopy temperature series (°C); $\bar{y_{i+k}}$ is the mean of atmospheric temperature series (°C); *R_m_* is the maximum correlation coefficient; *m* is the sliding shifts in the canopy temperature series that correspond to the maximum Pearson correlation coefficient; *k* = 0, ±1, ±2, …, ±*n*, *k* > 0 indicates the canopy temperature change ahead of atmospheric temperature; and *k* < 0 indicates that canopy temperature changes lag behind atmospheric temperature. *T_L_* is the time-lag parameter (min).

#### 3.4.5. Time-Lag Mutual Information Method

To date, no researcher has calculated the time-lag parameter between Tc and Ta using the time-lag mutual information method. Therefore, this study investigates it. Employing the time-lag mutual information method, the time-lag parameter between the canopy temperature, *X*, and atmospheric temperature, *Y*, is determined [37]. The formula is presented as follows:(8)$$I(X,Y,\tau )=\sum_{x}\sum_{y}p(x_{t},y_{t+\tau })\log\frac{p(x_{t},y_{t+\tau })}{p(x_{t},)p(y_{t+\tau })}$$
where $P(x_{t},y_{t+\tau })$ is the $X=x_{t},Y=y_{t+\tau }$ joint distribution probability. *τ* is the time-lag parameter. The time-lag parameter *τ* is determined when the mutual information coefficient reaches its peak. A positive *τ* means that *x* changes before *y*, while a negative *τ* indicates that *x* changes after *y*.

#### 3.4.6. Gray Time-Lag Correlation Analysis

Currently, no researcher has employed the gray the time-lag correlation analysis to investigate the time-lag effect between the Ta and Tc. Therefore, this study pioneers the use of gray time-lag correlation analysis to calculate the time-lag parameter between Ta and Tc. The methodology is outlined as follows:

① The reference sequence canopy temperature (Tc) is
(9)X=(x(1),x(2),…,x(n))

The comparison of the sequence group atmospheric temperature (Ta) is
(10)$$Y_{\tau }=(y(1+\tau ),y(2+\tau ),\ldots ,y(n+\tau ))$$
where *τ* is the time-lag parameter.

② Calculate the correlation coefficient $\zeta (x(k),y_{\tau }(k+\tau ))$ between *X* and *Y_τ_* with the following formula:(11)$$\zeta (x(k),y_{\tau }(k+\tau ))=\frac{\min\tau \min k\left|x(k)-y_{\tau }(k+\tau )\right|+\rho \max\tau \max k\left|x(k)-y_{\tau }(k+\tau )\right|}{\left|x(k)-y_{\tau }(k+\tau )\right|+\rho \max\tau \max k\left|x(k)-y_{\tau }(k+\tau )\right|}$$
(12)k=1,2,3,…,n
(13)τ=0,1,…,T−n
(14)$$\gamma (\tau )=\frac{1}{n}\sum_{k=1}^{n}\zeta (x(k),y_{\tau }(k+\tau ))$$
(15)τ=0,1,…,T−n
where *ρ* is the resolution factor, *ρ* = 0.5; *T* is the time span of the time series.

③The time lag parameter *τ* between Ta and Tc is identified as the time at which *γ*(*τ*) peaks.
(16)$$\gamma (\tau^{*})=\max_{0\leq \tau \leq T-n}\gamma (\tau )$$
where *γ*(*τ**) is the gray correlation between *X* and *Y*, $\tau^{*}$ is the time-lag parameter of *Y* and *X*.

#### 3.4.7. CWSI Theoretical Model

Based on the canopy energy balance theory, Jackson, Idso, Reginato and Pinter Jr [8] developed a theoretical model of the CWSI. The formula is as follows:(17)CWSI=γ(1+rcra)−γ*Δ+γ(1+rcra)
(18)$$\gamma =0.665\times 101.3\times {\left(\frac{293-0.0065Z}{293}\right)}^{5.26}$$
(19)γ*=γ×(1+rcra)
(20)$$\Delta =45.03+3.014T+0.05345T^{2}+0.00224T^{3}$$
(21)$$T=\frac{Tc+Ta}{2}$$
(22)$$r_{a}=\frac{4.72{\left[\ln(\frac{z-d}{z_{0}})\right]}^{2}}{\left(1+0.54u\right)}$$
where the *CWSI* is crop water stress index; *γ* is psychrometric coefficient (Pa·°C^−1^); *r_c_* is canopy resistance (s·m^−1^); *r_a_* is aerodynamics resistance (s·m^−1^); Δ is slope of the water vapor pressure curve (Pa·°C^−1^); *Z* is height above sea level (m); *d* is zero-plane displacement (m), *d* = 0.63 h; *z*_0_ is roughness (m), *z*_0_ = 0.13 h; *h* is crop height (m); *u* is reference height wind speed (m·s^−1^); *z* is reference height (m), *z* = 2; and *r_c_* is canopy resistance (s·m^−1^), displayed in Table 4.

#### 3.4.8. CWSI Empirical Model

*The CWSI* empirical model was first constructed by Idso et al. [7]. The formula is as follows:(23)$$CWSI=\frac{(Tc-Ta)-NWSB}{NTB-NWSB}$$
(24)CTD=Tc−Ta
(25)$$VPD=0.6108\times \exp(\frac{17.27\times Ta}{Ta+237.7})\times (\frac{100-RH}{100})$$
(26)$$VPG=0.6108\times \exp(\frac{17.27\times Ta}{Ta\times 237.7})-0.6108\times \exp(\frac{17.27\times (Ta+b)}{(Ta+b)+237.7})$$
(27)NWSB=a×VPD+b
(28)NTB=a×VPG+b
where *Tc* is canopy temperature (°C); *Ta* is atmospheric temperature (°C); *NWSB* is lower bound (no water stress); *NTB* is upper bound (no transpiration); *CTD* is canopy air temperature differential (°C); and *a*, *b* are the slope and intercept of CTD and VPD linear fits, respectively.

Solar radiation intensifies during the period from 13:00 to 15:00, when the discrepancy between crop and soil water supply conditions becomes more pronounced, and the linear relationship between the canopy temperature difference (CTD) and vapor pressure deficit (VPD) is distinct [39]. Consequently, this study opts for a linear fitting of the CTD and VPD specifically for the 13:00–15:00 interval. The results are presented in Figure 29.

#### 3.4.9. Evaluation Indicators

In this study, the accuracy of the CWSI inversion of photosynthetic parameters, both before and after time-lag corrections, is assessed using the coefficient of determination (R^2^). An R^2^ value closer to 1 indicates a higher inversion accuracy.

Similarly, the prediction accuracy of the machine learning algorithm is evaluated through the coefficient of determination (R^2^) and the root-mean-square error (RMSE), with R^2^ values nearing 1 and RMSE values approaching 0 denoting enhanced prediction accuracy.

#### 3.4.10. Machine Learning Algorithms

In this study, various machine learning and deep learning methods were employed to process and predict the crop water stress index (CWSI), including Genetic Algorithm Optimized Support Vector Machines (GA-SVMs), Bayesian Optimized Long Short-Term Memory Neural Networks (Bayes-LSTMs), Particle Swarm Algorithm Optimized Long Short-Term Memory (PSO-LSTM), Convolutional Bi-directional Long Short-Term Memory Neural Networks (CNN-BILSTMs), Attention Mechanism Long Short-Term Memory Neural Networks (Attention-LSTMs), and Attention Mechanism Gated Recurrent Units (Attention-GRUs). The GA-SVM optimizes SVM parameters using a genetic algorithm, effectively enhancing the model’s classification and prediction performance, making it suitable for small but complex datasets. PSO-LSTM employs particle swarm optimization to find the optimal parameters for LSTM, improving prediction performance and training efficiency, suitable for scenarios with a large parameter space. The CNN-BILSTM combines a CNN and bi-directional LSTM to simultaneously extract spatial and temporal features, enhancing the prediction capability for complex long time-series data with spatial dependencies. The Attention-LSTM incorporates an attention mechanism into LSTM, enhancing the model’s focus on important time steps and improving prediction accuracy, particularly for long time-series data with significant features. The Attention-GRU introduces an attention mechanism into the GRU, simplifying the network structure while improving the focus on important time steps, making it suitable for the efficient prediction of long time-series data. Overall, the introduction of the attention mechanisms (Attention-LSTM and Attention-GRU) significantly enhances the model’s ability to capture important information, thus improving prediction accuracy. Bayes-LSTM enhances model robustness by addressing parameter uncertainty. Both PSO-LSTM and GA-SVM improve model performance through optimization algorithms, but are sensitive to initial settings and optimization processes.

## 4. Discussions

### 4.1. Time-Lag Parameters between the Tc and Ta Calculated by Different Models

The essence of the peak-finding method is to find a suitable function for fitting [40], and the time difference of the peak of the curve is the time-lag parameter between the two series. In this study, the CCE equation was applied to fit the Tc of winter wheat after S-G filter smoothing, and the ECS equation was applied to fit the Ta after S-G filter smoothing. This is in general agreement with the time-lag parameter between the Tc and Ta for summer maize obtained by Zhang et al. [34]. Considering that the time-lag effect was the persistent influence of previous climatic conditions on current crop growth as a result of the cumulative effects of meteorological factors and soil moisture content on the crop [41,42], there were limitations in determining the time-lag parameter through isolated points. Therefore, the time-lag parameter between the Ta and Tc was calculated using the time-lag cross-correlation method [43]. The time-lag parameter calculated in this study was about 32–44 min, consistent with the findings of Zhang et al. [44].

Meanwhile, this study innovatively utilized the time-lag mutual information method [45] and gray time-lag correlation analysis [46] to calculate the time-lag parameter between the Ta and Tc. The time-lag parameter calculated by the time-lag mutual information method ranged from 42 to 58 min, while the gray time-lag correlation analysis-calculated time-lag parameter ranged from 76 to 98 min. Additionally, the time-lag parameter between the Tc and Ta in winter wheat calculated by the four methods all experienced a significant sudden drop under the heavy water stress treatment. Pn, Tr, and gs all exhibited a decreasing trend with diminishing soil moisture [47], and a sudden drop occurred during the severe water stress treatment (T4) [2]. This phenomenon might be related to the soil moisture quench value [5].

Mild water stress does not affect the normal life activities of the crop, and the physiological activities of the plant are limited only when the degree of drought stress exceeds the drought threshold. When the soil moisture threshold is reached, stomata are reduced or closed, and water lost through stomatal transpiration and CO_2_ entering the chloroplasts is reduced [48]. As a result, Pn, Tr, and gs undergo varying degrees of reduction [49]. Wu et al. [50] found that, when the soil volumetric water content was lower than 60% of the field holding capacity for a long period of time, leaf enlargement was restricted, the total leaf area for light energy interception was reduced, and the gas exchange process of winter wheat was limited, which was the reason for the sudden decrease in photosynthetic parameters under severe water stress. At the same time, the decrease in stomatal conductance reduces crop transpiration, evaporative cooling was reduced, and canopy temperatures continue to rise [51], reaching their peaks later, resulting in a decrease in the time-lag parameter between the Tc and Ta in the heavy water stress treatment. Liu et al. [52] found that the soil moisture quench value of winter wheat was about 43.5–52.2% of the soil water content, which was consistent with the soil moisture treatments in this experiment, where there were abrupt changes in photosynthetic and time-lag parameters.

### 4.2. Reasons for Different Changes in the Magnitude of the Accuracy of the CWSI Inversions of Pn, Tr, and gs before and after Corrections of the Time-Lag Effect

After considering the time-lag effect, the magnitudes of correlations between the CWSI and Pn, Tr, and gs varied inconsistently, which might be related to the different major environmental factors affecting Pn, Tr, and gs [29], as well as their distinct critical soil moisture thresholds. When the crop was not subjected to water stress, environmental factors had a small and negligible effect on Pn, Tr was mainly limited by solar radiation, and gs was primarily limited by photosynthetically active radiation and crop canopy temperature. When crops were subjected to water stress, Pn was mainly limited by relative humidity and atmospheric temperature, Tr was chiefly limited by saturated water-vapor pressure difference, and gs was predominantly limited by saturated water-vapor pressure difference and wind speed [53]. Meanwhile, Pn, Tr, and gs showed different sensitivities to soil water deficit. The critical soil moisture thresholds for Pn, Tr, and gs were 62%, 60%, and 58% for maize at the seedling stage and 51%, 53%, and 48% at the nodulation stage, respectively. This indicated that crop photosynthetic parameters were sensitive to soil moisture in the order of gs > Tr > Pn [14], consistent with the magnitude of the CWSI correlation with Pn, Tr, and gs obtained in this study [18]. At the same time, this might result in a varying degree of improvement in the correlation between the CWSI and Pn, Tr, and gs before and after accounting for time-lag effects (gs > Tr > Pn).

### 4.3. Outlook

Physiological parameters of plants at different growth stages exhibit varying sensitivities to soil moisture [14]. This suggests that the photosynthetic parameters of winter wheat at different fertility stages show differential sensitivities to the CWSI under varying water stress conditions. The impact of water stress on the crop’s gas exchange processes is minimal during the regrowth period, with little variation in Pn, Tr, and Gs across different water stress levels. However, the inhibitory effects of persistent water stress during the nodulation–irrigation period are more pronounced, indicating a more significant decrease in Pn, Tr, and gs in winter wheat under severe water stress [54]. Therefore, there are significant seasonal variations in the correlation of the CWSI with Pn, Tr, and gs in winter wheat subjected to different moisture treatments. In this study, the impact of the time-lag effect on the accuracy of the CWSI inversion of photosynthesis parameters is investigated only for the entire reproductive period. The influence of time lag between the Ta and Tc on the accuracy of the CWSI inversion of photosynthesis parameters during each reproductive phase is not discussed and requires further study.

## 5. Conclusions

In this study, we investigate the impact of the time-lag effect between the Tc and Ta on the correlation between the CWSI and photosynthetic parameters. The main conclusions are: (1) The magnitude of the time-lag parameter between the Tc and Ta in winter wheat, calculated by the four methods for the entire reproductive period, follows the order: gray time-lag correlation analysis > time-lag peak-seeking method > time-lag mutual information method > time-lag cross-correlation method. All time-lag parameters of severe water stress treatment experience a sudden decrease. (2) The CWSI empirical model is more sensitive to the time-lag effect than the theoretical model. Time-lag correction, particularly using the time-lag mutual information method, significantly improves the correlation between the CWSI and photosynthetic parameters. (3) The GA-SVM machine learning algorithm provides the highest prediction accuracy for daily changes in the CWSI empirical model corrected with the time-lag mutual information method (R^2^ = 0.982, RMSE = 0.017).

## Figures and Tables

**Figure 1 plants-13-01702-f001:**
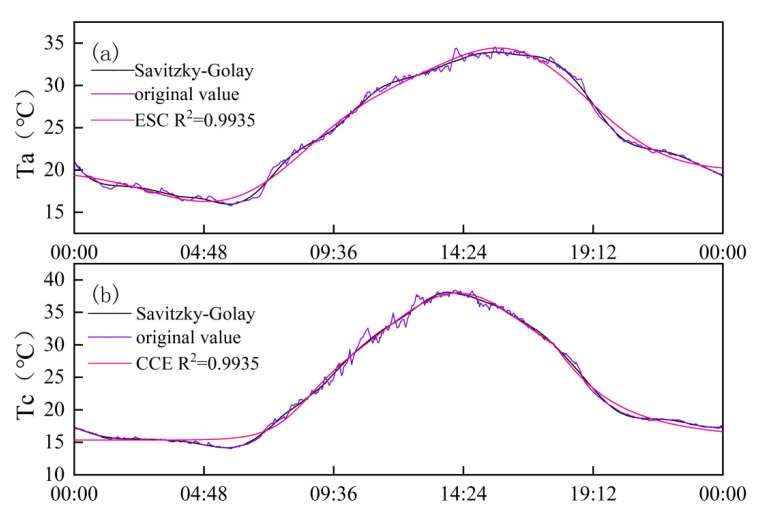
(**a**) ESC equations fitted to S-G filter-smoothed Ta; (**b**) CCE equation fitted to S-G filter-smoothed Tc.

**Figure 2 plants-13-01702-f002:**
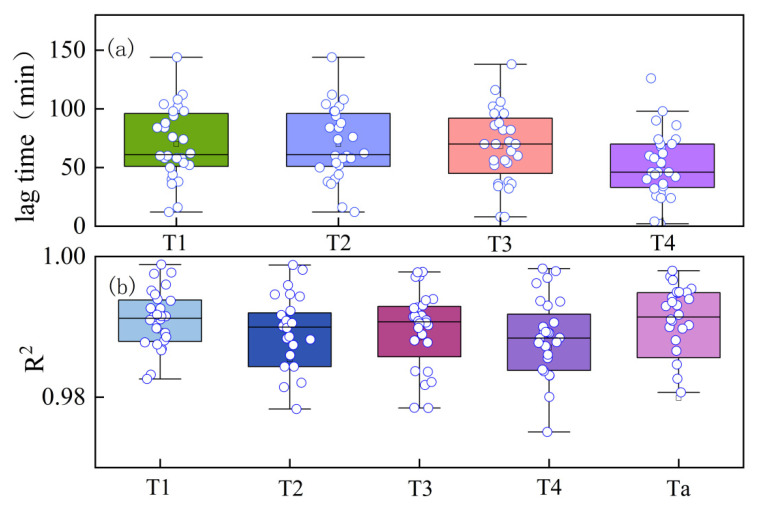
(**a**) Time-lag parameters of T1 (fully irrigated), T2 (mild water stress), T3 (moderate water stress), and T4 (severe water stress) calculated by time-lag peak-finding method. (**b**) Coefficient of determination (R^2^) for T1, T2, T3, and T4 fitted by the time-lag peak-finding method.

**Figure 3 plants-13-01702-f003:**
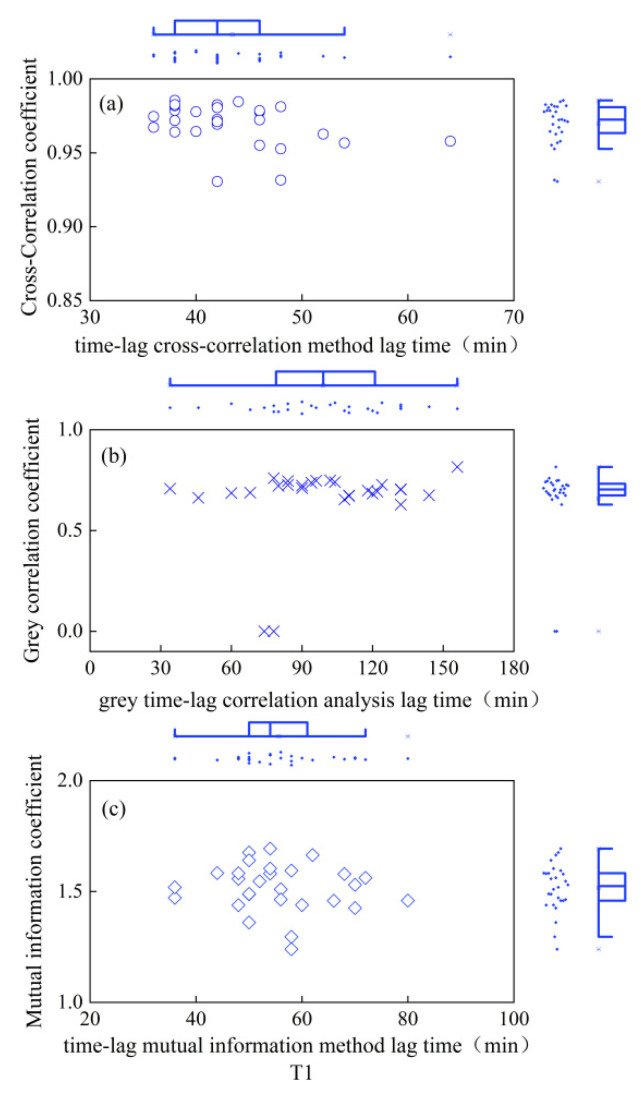
Time-lag parameters and corresponding coefficients for fully irrigated treatment. (**a**) is the time-lag parameter calculated by the time-lag cross-correlation method and the cross-correlation coefficient between Ta and Tc after the corrected time-lag; (**b**) is the time-lag parameter calculated by the time-lag grey correlation analysis and the time-lag grey correlation coefficient between Ta and Tc after the corrected time-lag; (**c**) is the time-lag parameter calculated by the time-lag mutual information method and the mutual information coefficient between Ta and Tc after the corrected time-lag. Circles indicate the results of the time-lag cross-correlation method under the four moisture treatments; cross sign indicates the results of the time-lag grey correlation analysis under the four moisture treatments; squares indicate the results of the time- lag mutual information method.

**Figure 4 plants-13-01702-f004:**
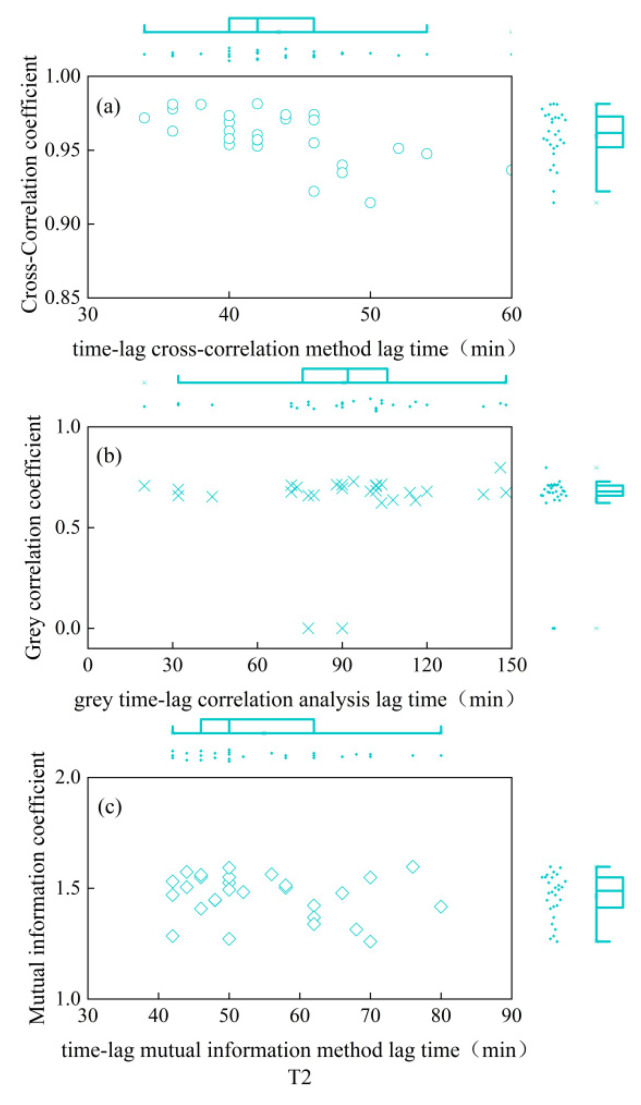
Time-lag parameters and corresponding coefficients for mild water stress treatment. (**a**) is the time-lag parameter calculated by the time-lag cross-correlation method and the cross-correlation coefficient between Ta and Tc after the corrected time-lag; (**b**) is the time-lag parameter calculated by the time-lag grey correlation analysis and the time-lag grey correlation coefficient between Ta and Tc after the corrected time-lag; (**c**) is the time-lag parameter calculated by the time-lag mutual information method and the mutual information coefficient between Ta and Tc after the corrected time-lag. Circles indicate the results of the time-lag cross-correlation method under the four moisture treatments; cross sign indicates the results of the time-lag grey correlation analysis under the four moisture treatments; squares indicate the results of the time- lag mutual information method.

**Figure 5 plants-13-01702-f005:**
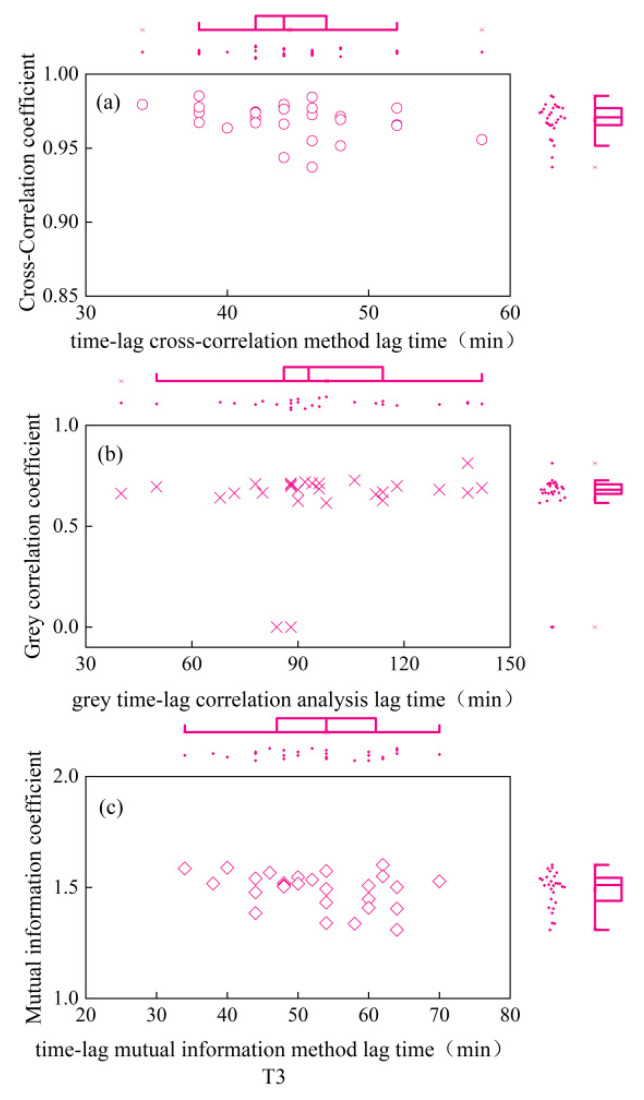
Time-lag parameters and corresponding coefficients for moderate water stress treatment. (**a**) is the time-lag parameter calculated by the time-lag cross-correlation method and the cross-correlation coefficient between Ta and Tc after the corrected time-lag; (**b**) is the time-lag parameter calculated by the time-lag grey correlation analysis and the time-lag grey correlation coefficient between Ta and Tc after the corrected time-lag; (**c**) is the time-lag parameter calculated by the time-lag mutual information method and the mutual information coefficient between Ta and Tc after the corrected time-lag. Circles indicate the results of the time-lag cross-correlation method under the four moisture treatments; cross sign indicates the results of the time-lag grey correlation analysis under the four moisture treatments; squares indicate the results of the time- lag mutual information method.

**Figure 6 plants-13-01702-f006:**
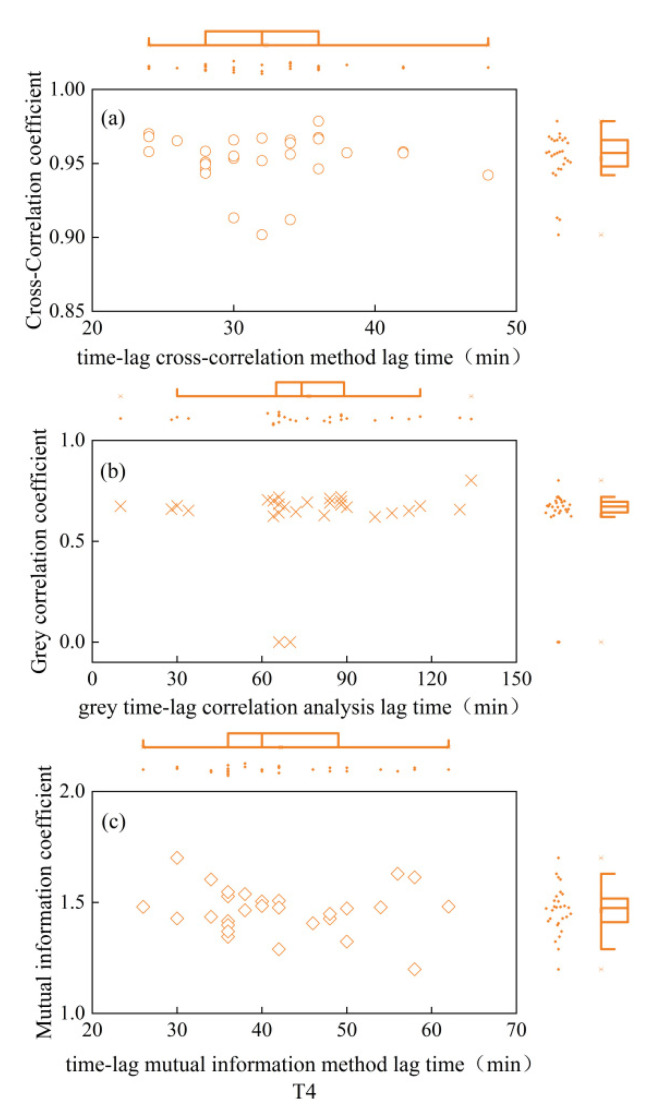
Time-lag parameters and corresponding coefficients for severe water stress treatment. (**a**) is the time-lag parameter calculated by the time-lag cross-correlation method and the cross-correlation coefficient between Ta and Tc after the corrected time-lag; (**b**) is the time-lag parameter calculated by the time-lag grey correlation analysis and the time-lag grey correlation coefficient between Ta and Tc after the corrected time-lag; (**c**) is the time-lag parameter calculated by the time-lag mutual information method and the mutual information coefficient between Ta and Tc after the corrected time-lag. Circles indicate the results of the time-lag cross-correlation method under the four moisture treatments; cross sign indicates the results of the time-lag grey correlation analysis under the four moisture treatments; squares indicate the results of the time- lag mutual information method.

**Figure 7 plants-13-01702-f007:**
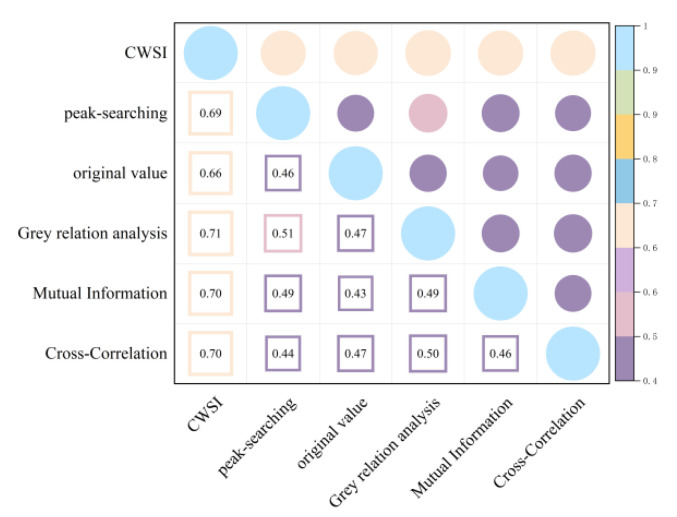
Heat map of the CWSI theoretical model and Pn before and after considering time-lag effects.

**Figure 8 plants-13-01702-f008:**
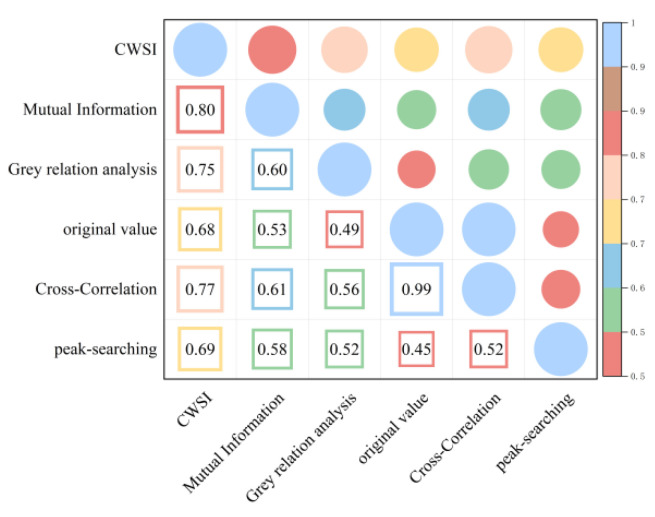
Heat map of the CWSI empirical model and Pn before and after considering time-lag effects.

**Figure 9 plants-13-01702-f009:**
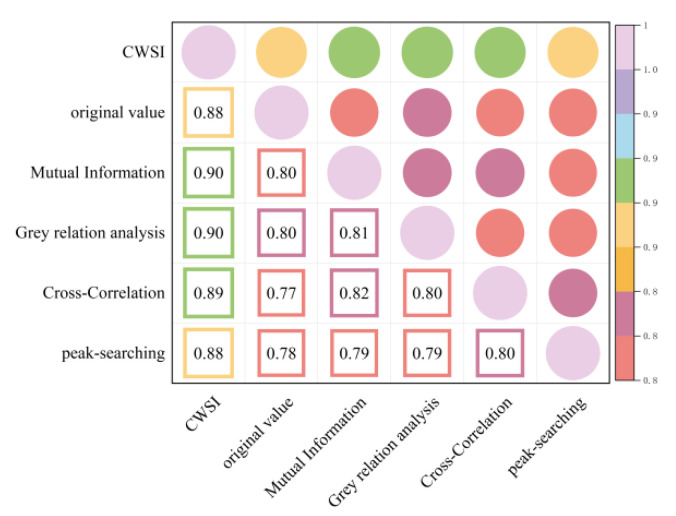
Heat map of the CWSI theoretical model and Tr before and after considering time-lag effects.

**Figure 10 plants-13-01702-f010:**
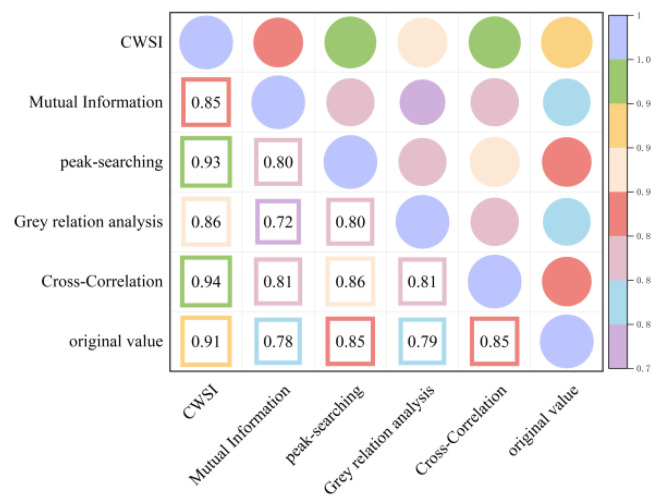
Heat map of the CWSI empirical model and Tr before and after considering time-lag effects.

**Figure 11 plants-13-01702-f011:**
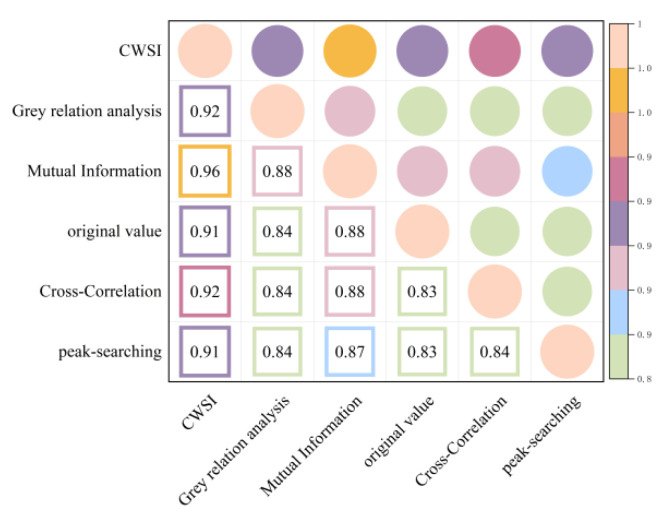
Heat map of the CWSI theoretical model and gs before and after considering time-lag effects.

**Figure 12 plants-13-01702-f012:**
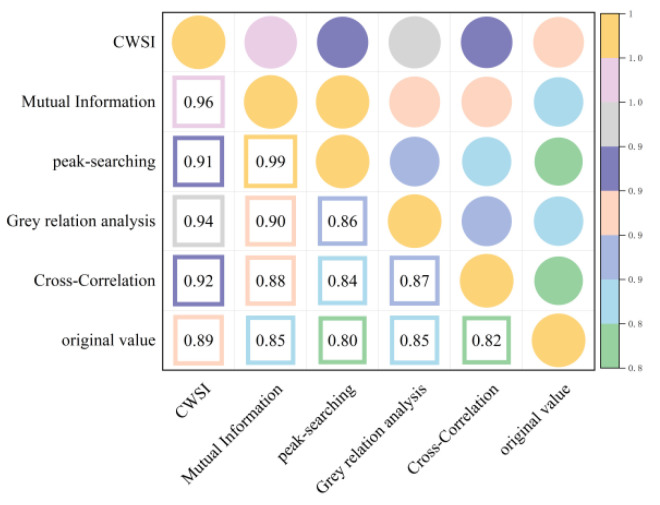
Heat map of the CWSI empirical model and gs before and after considering time-lag effects.

**Figure 13 plants-13-01702-f013:**
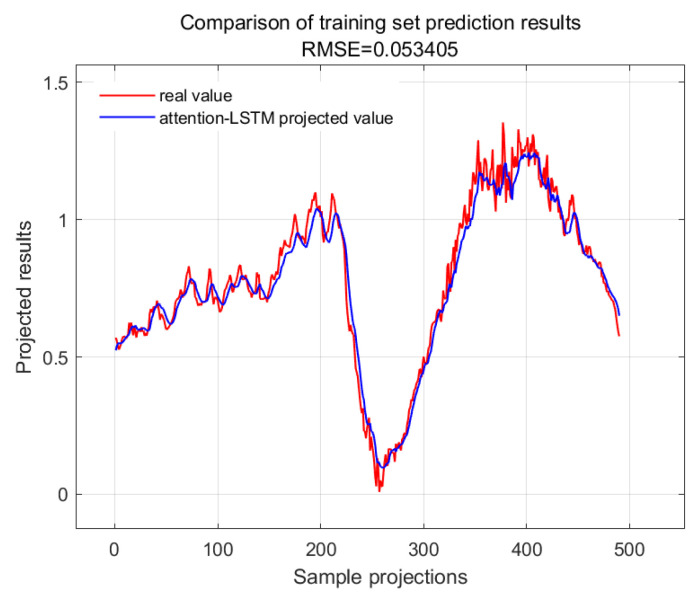
Training set for attention-LSTM.

**Figure 14 plants-13-01702-f014:**
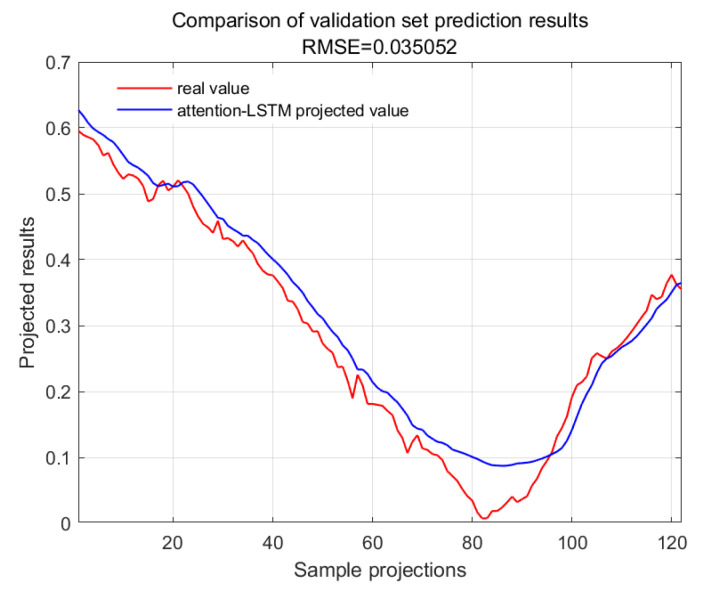
Validation set for attention-LSTM.

**Figure 15 plants-13-01702-f015:**
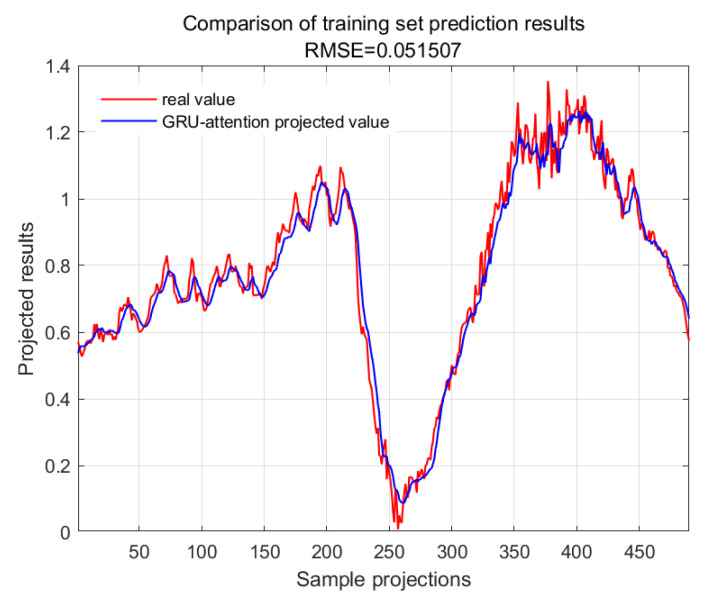
Training set for GRU-attention.

**Figure 16 plants-13-01702-f016:**
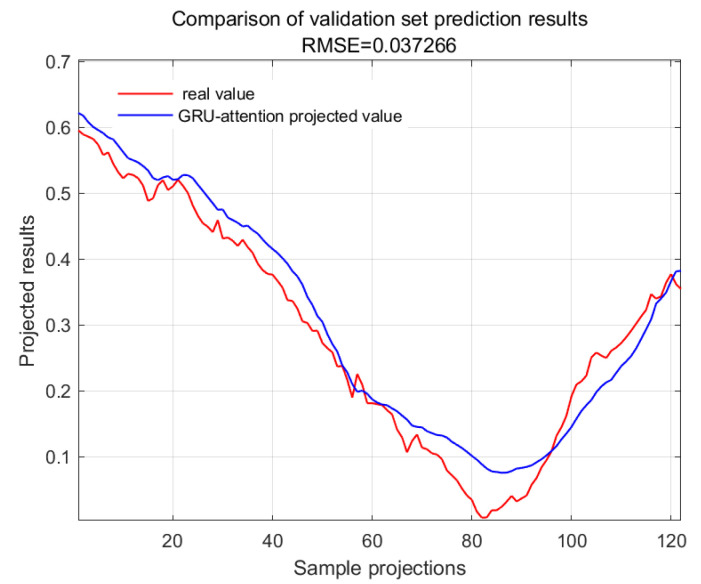
Validation set for GRU-attention.

**Figure 17 plants-13-01702-f017:**
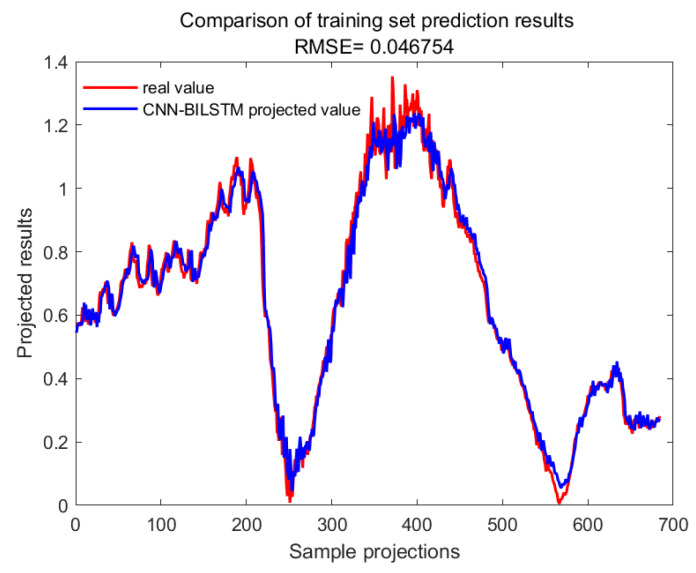
Training set for CNN-BILSTM.

**Figure 18 plants-13-01702-f018:**
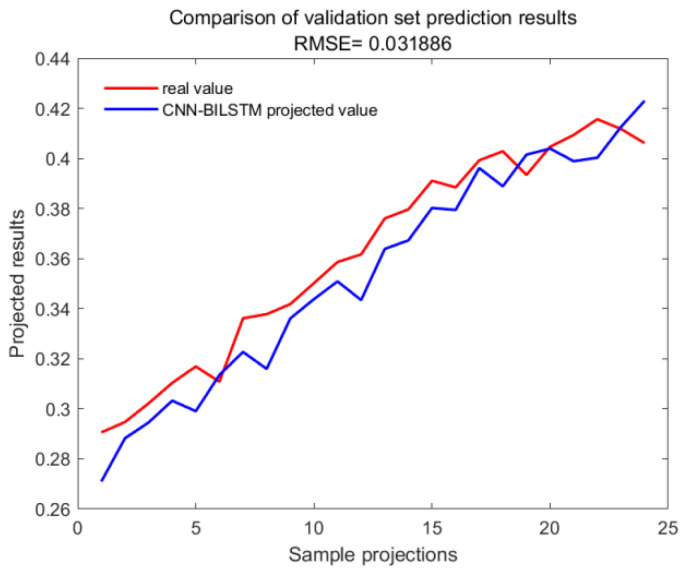
Validation set for CNN-BILSTM.

**Figure 19 plants-13-01702-f019:**
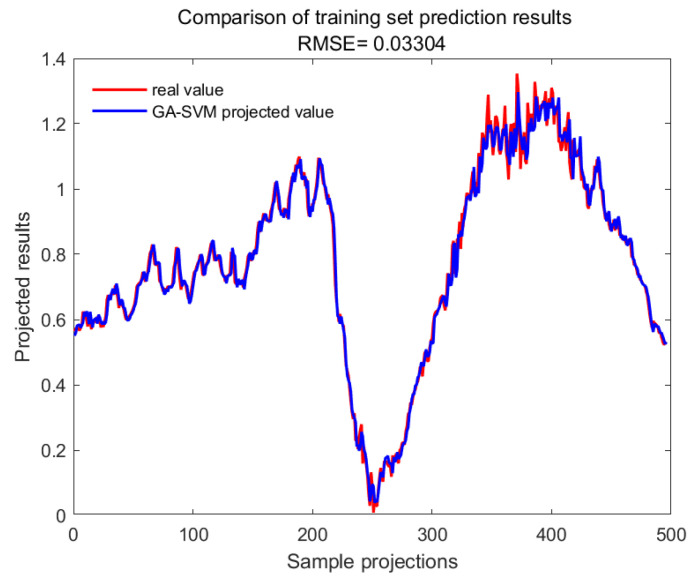
Training set for GA-SVM.

**Figure 20 plants-13-01702-f020:**
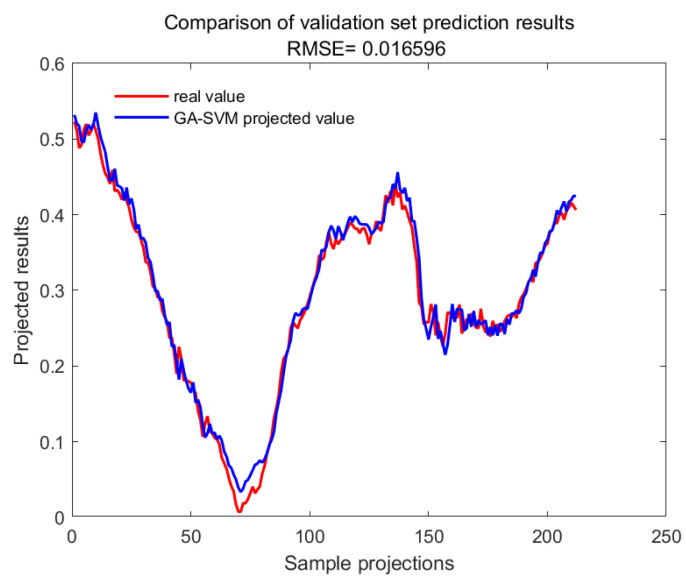
Validation set for GA-SVM.

**Figure 21 plants-13-01702-f021:**
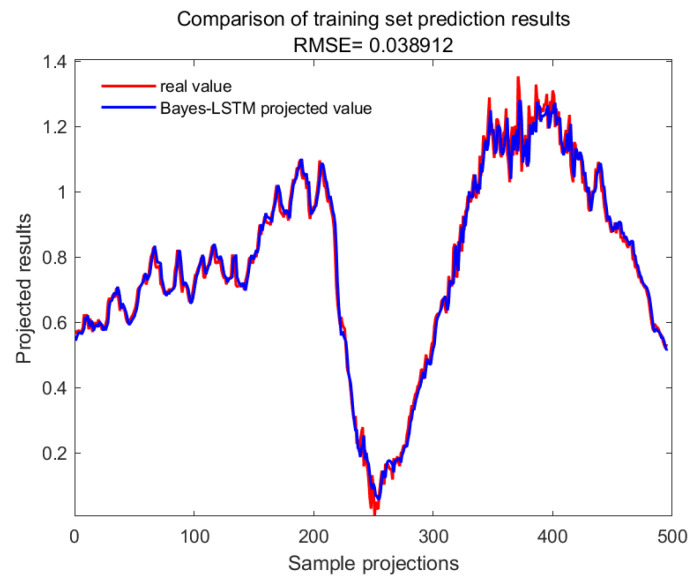
Training set for Bayes-LSTM.

**Figure 22 plants-13-01702-f022:**
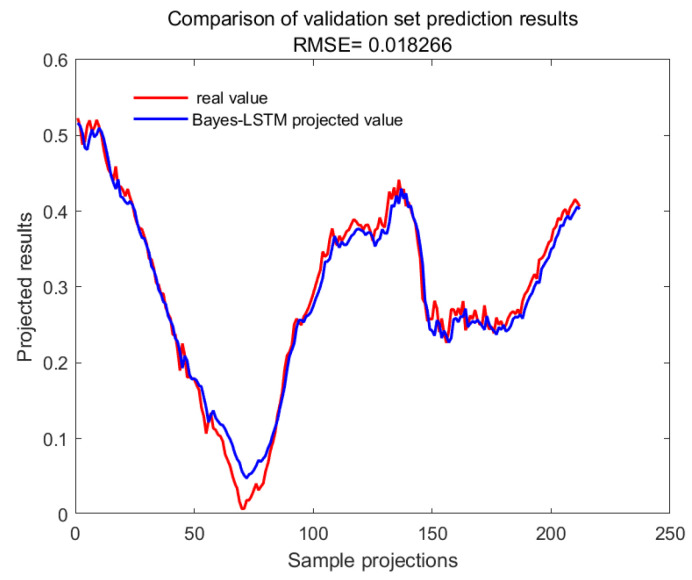
Validation set for Bayes-LSTM.

**Figure 23 plants-13-01702-f023:**
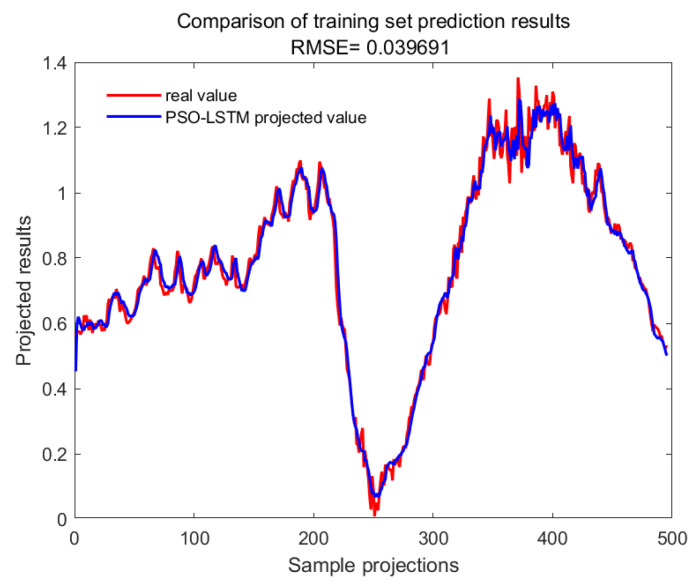
Training set for PSO-LSTM.

**Figure 24 plants-13-01702-f024:**
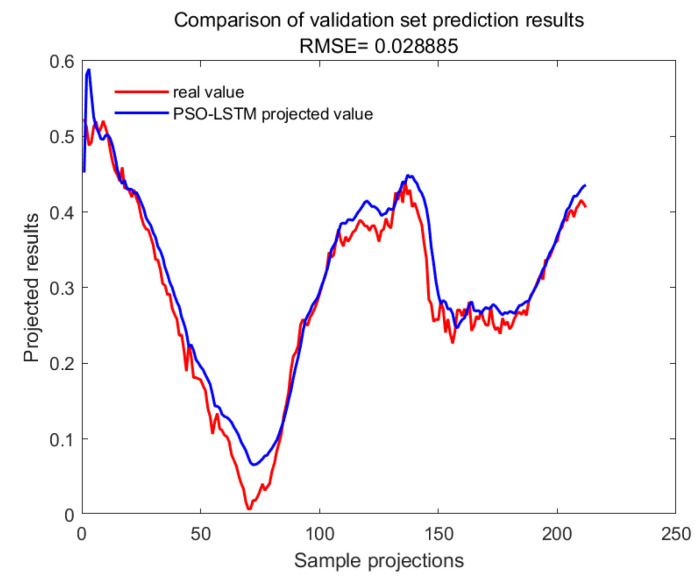
Validation set for PSO-LSTM.

**Figure 25 plants-13-01702-f025:**
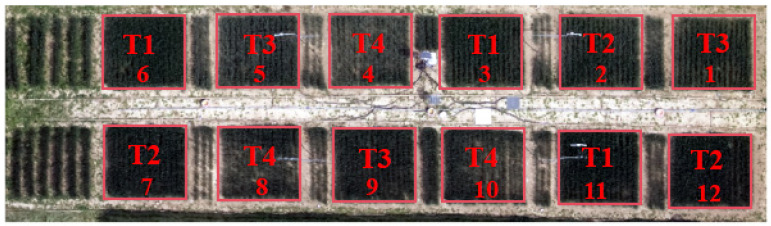
Overview of the experimental site.

**Figure 26 plants-13-01702-f026:**
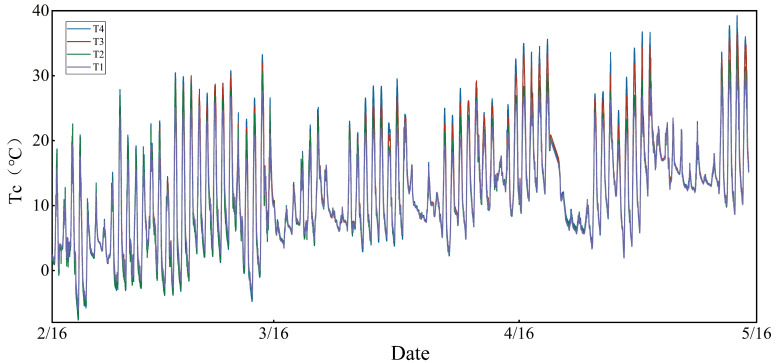
Canopy temperature of winter wheat under four moisture treatments.

**Figure 27 plants-13-01702-f027:**
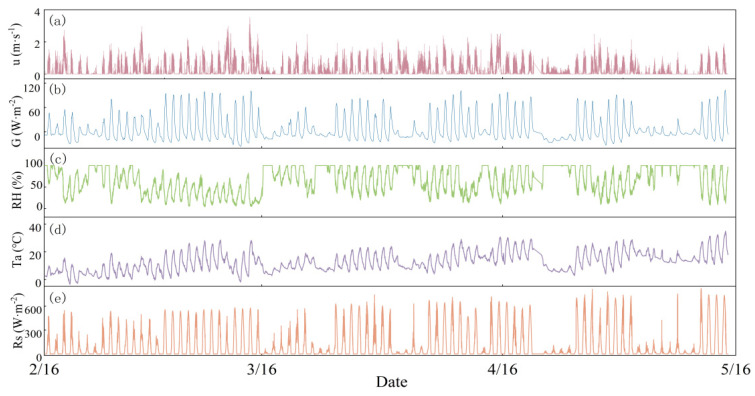
Dynamic variations in (**a**) u (m·s^−1^); (**b**) G (W·m^−2^); (**c**) RH (%); (**d**) Ta (°C); (**e**) Rs (W·m^−2^).

**Figure 28 plants-13-01702-f028:**
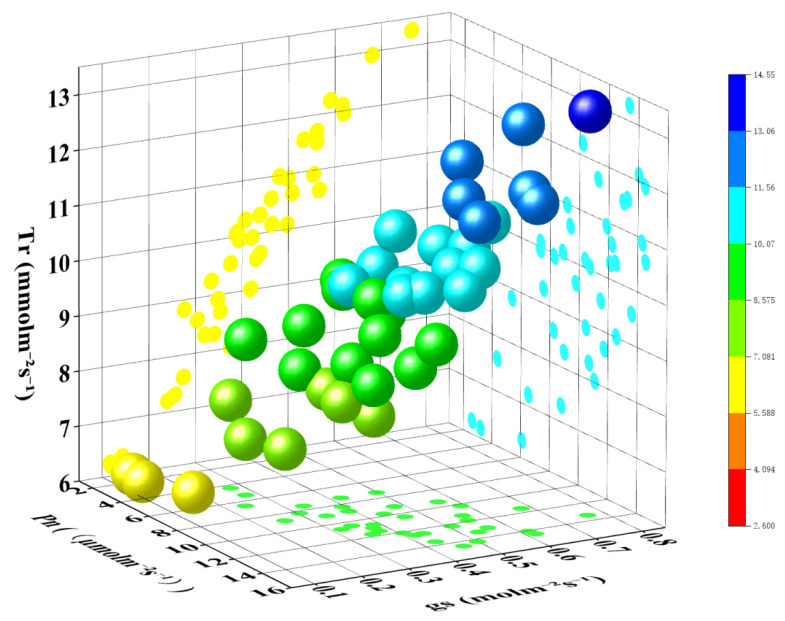
Pn (µmol/(m^2^·s)), gs (mol/(m^2^·s)), and Tr (mmol/(m^2^·s)) in winter wheat.

**Figure 29 plants-13-01702-f029:**
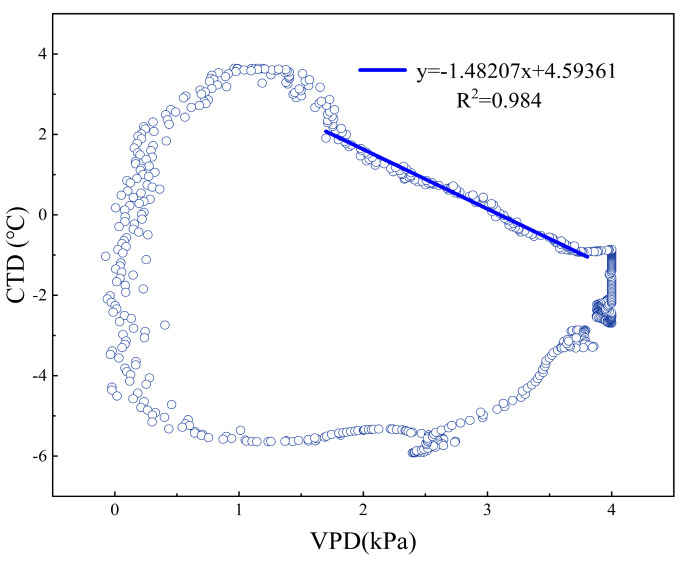
The blue circle represents the trajectory of the VPD (kPa) and the corresponding CTD (°C) over the course of a day; The blue line represents the outcome of a linear regression analysis of the relationship between VPD (kPa) and CTD (°C) between 13:00 and 15:00.

**Table 1 plants-13-01702-t001:** Prediction accuracy of machine learning algorithms for the CWSI empirical model based on time-lag mutual information correction.

Machine Learning Algorithm	R^2^	RMSE
attention-LSTM	0.88928	0.035052
GRU-attention	0.80197	0.037266
CNN-BILSTM	0.9148	0.031886
GA-SVM	0.98237	0.016596
PSO-LSTM	0.9466	0.028885
Bayes-LSTM	0.97865	0.018266

**Table 2 plants-13-01702-t002:** Irrigation quota of winter wheat.

Irrigation Date	Irrigation Quota (mm)			
	T1	T2	T3	T4
17 February 2023	46.7	43.9	19.2	18.2
25 February 2023	50.3	23.8	23.0	16.0
5 March 2023	63.0	28.5	13.4	31.2
12 March 2023	58.1	19.4	17.7	51.0
19 March 2023	64.0	54.6	40.0	16.3
29 March 2023	64.0	44.8	33.0	16.3
7 April 2023	90.3	36.0	15.8	38.0
18 April 2023	91.9	44.9	40.6	42.0
28 April 2023	29.5	25.2	19.4	17.0
12 May 2023	53.8	69.0	88.4	14.0
23 May 2023	53.9	65.0	30.6	17.9

**Table 3 plants-13-01702-t003:** Summary of environmental factors observed by the weather station.

Variables	Sensor Number	Instrument Height (m)	Abbreviation	Unit
Solar radiation	SN-500	3.5	Rs	W·m^−2^
Soil heat fux	HFP01	−0.10	G	W·m^−2^
Atmospheric temperature	HC2AS3	2.5	Ta	°C
Relative humidity	HC2AS3	2.5	RH	%
Wind speed	HC2AS3	2	u	m·s^−1^

**Table 4 plants-13-01702-t004:** r_c_ of winter wheat at different fertility stages [38].

Growth Period	r_c_ (s·m^−1^)
Regreening stage-jointing stage	13.01
Jointing stage-tasseling stage	18.03
Tasseling stage-filling stage	26.85

## Data Availability

Data are contained within the article.

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
