# Peer review of "Influence of Time-Lag Effects between Winter-Wheat Canopy Temperature and Atmospheric Temperature on the Accuracy of CWSI Inversion of Photosynthetic Parameters"

_plants, 2024, doi:10.3390/plants13121702_

Round 1
Reviewer 1 Report
Comments and Suggestions for Authors
The article Influence of time lag effect between winter wheat canopy temperature and atmospheric temperature on the accuracy of CWSI inversion of photosynthetic parameters quantified the time-lag parameter in winter wheat. The authors applied several machine learning algorithms to predict the daily variation of CWSI after time-lag correction. I believe it is worth publishing in the current journal, but I have a few comments to improve it.
1) Abstract:
1.1 It is necessary to make clear the aim of the research.
2) Introduction:
2.1 The third paragraph is too short.
2.2 The hypothesis is missing.
3) Material and methods
I would recommend to explain if the assumptions of the models that have been used have been attempt.
4) Results
Instead of using abbreviations (e.g. T1, T2, T3…), I suggest using the complete name of the treatments.
All symbols and abbreviations need to be defined in the legend of the figures and tables.
5) Conclusion
It needs to be figured out. Please use the present tense and state the most important outcome of your work.
Author Response
Dear editor and reviewer,
Thank you for offering us an opportunity to improve the quality of our submitted manuscript (Influence of time lag effects between winter wheat canopy temperature and atmospheric temperature on the accuracy of CWSI inversion of photosynthetic parameters, manuscript mumber: 3023810). The time and efforts you spent to suggest improvements to our paper are much appreciated. Thank you for your careful review, We have incorporated your comments and prepared the revision for your review.The comments of the editor and reviewers are addressed on an individual basis. We hope that the responses would meet with your approval. Changes made in revised manuscript are highlighted with yellow colour. Please find our responses to each of the comments point by point below.
Abstract:
Q1: 1.1It is necessary to make clear the aim of the research.
Response:Thanks for your valuable suggestion.We have added the following content to the abstract:"When calculating CWSI, previous researchers usually used canopy temperature and atmospheric temperature at the same time. However, it takes some time for canopy temperature (Tc) to respond to atmospheric temperature(Ta), suggesting the time lag effects between Ta and Tc. In order to investigate time lag effects between Ta and Tc on the accuracy of CWSI inversion of photosynthetic parameters in winter wheat, we conducted the experiment. "
Introduction:
Q2: 2.1The third paragraph is too short.
Response:Thanks for your valuable suggestion. Here's the third paragraph of the content we added:"CWSI is a sensitive indicator to reflect the water stress caused by stomatal function of the crop, and continuous water stress results in an increasing trend of CWSI and a decreasing trend of Pn、Tr and gs[9]. There is a good negative correlation between CWSI and photosynthetic parameters [6,10].The results of Ramos-Fernández, et al. [11]showed a strong correlation between CWSI and gs(R2 = 0.91). When the crop is subjected to water stress, the soil-root hydraulic resistance increases [12], which reduces root water transport and eventually leads to the reduction or closure of plant stomata and a decrease in photosynthetic parameters[13]. Different physiological characteristics of wheat have different sensitivities to soil moisture [14], therefore the correlation between Pn, gs, Tr and CWSI varies."
Q3: 2.2The hypothesis is missing.
Response:Thanks for your valuable suggestion.We made assumptions in the fifth paragraph, which are as follows:"We hypothesised that the time lag effects between canopy temperature and atmospheric temperature has a significant impact on the model accuracy of the CWSI inversion of photosynthetic parameters.Therefore, we conducted an experiment with winter wheat where we continuously monitored canopy temperature and environmental factors of winter wheat. We quantified the time-lag parameters between Ta and Tc using time-lag peak-finding, time-lag cross-correlation, time-lag mutual information and time-lag grey correlation analysis. We then modified the theoretical and empirical CWSI models based on these time-lag parameters. Finally, we investigated the implication and mechanisms of Ta and Tc time-lag effects on the accuracy of CWSI inversion of photosynthesis parameters."
Material and methods
Q3:I would recommend to explain if the assumptions of the models that have been used have been attempt.
Response:Thanks for your valuable suggestion.
We supplemented the 2.4.3. Time-lag peak-seeking method section as follows:"Zhang and Wu [24] used the Gaussian function to fit the canopy temperature and atmospheric temperature of summer maize and achieved good accuracy. However, the Gaussian function fits the canopy temperature and atmospheric temperature of winter wheat with lower accuracy ";
We supplemented the 2.4.4.Time-lag cross-correlation method section as follows:"Zhang, et al. [25]used the time-lag cross-correlation method to calculate the time lag between canopy temperature and atmospheric temperature in winter wheat. They then found that correcting the time-lag effect between Tc and Ta by the time-lag cross-correlation method can improve the accuracy of CWSI inversion of SWC";
We supplemented the 2.4.5. Time-lag mutual information method section as follows:"To date, no researcher has calculated the time lag parameter of Tc and Ta using the time lag mutual information method. Therefore, this study investigates it";
We supplemented the 2.4.6. Grey time-lag correlation analysis section as follows:"Currently, no researcher has employed grey time-lag correlation analysis to investigate the time lag effect between Ta and Tc. Therefore, this study pioneers the use of grey time-lag correlation analysis to calculate the time lag parameter between Ta and Tc. The methodology is outlined as follows:";
Results
Q4: Instead of using abbreviations (e.g. T1, T2, T3…), I suggest using the complete name of the treatments.
Response:Thanks for your valuable suggestion."We made modifications to the second, third, and fourth paragraphs of section 3.1. Time-lag parameters of winter wheat under different water stresses. The modified content is as follows:
"The time lag parameters between canopy temperature (Tc) and atmospheric temperature (Ta), calculated using four different methods, exhibited distinct values across varying irrigation treatments. For the fully irrigated treatment, the time lag parameters were approximately 53 minutes, 44 minutes, 58 minutes, and 97 minutes when calculated using the time-lag peak-finding method, time-lag cross-correlation method, time-lag mutual information method, and grey time-lag correlation analysis, respectively. For the mild water stress treatment, these time lag parameters were about 52 minutes, 43 minutes, 55 minutes, and 92 minutes, respectively. For the moderate water stress treatment, the parameters were approximately 55 minutes, 44 minutes, 54 minutes, and 98 minutes. Lastly, for the severe water stress treatment, the parameters were around 44 minutes, 32 minutes, 42 minutes, and 76 minutes. These results highlight the variability in time lag parameters across different irrigation treatments, as well as the influence of the chosen calculation method.
This indicates that the time lag between Tc and Ta obtained from different calculation methods for the fully irrigated, mild water stress, and moderate water stress treatments did not differ significantly. However, for the severe water stress treatment, Tc reached its peak time later, resulting in a decrease in the time lag parameter between Ta and Tc by approximately 10 to 22 minutes. This phenomenon might be associated with the soil moisture threshold [31]. When the soil moisture threshold was reached, the water lost through transpiration in winter wheat could not be replenished promptly. To ensure the normal life activities of the crop, the expansion rate of the crop leaves was reduced, stomatal conductance decreased significantly, transpiration rate declined, and canopy temperature continued to increase, reaching the peak time later. This led to a shorter time lag between atmospheric temperature and canopy temperature [32].
In addition, the cross-correlation coefficient, mutual information coefficient, and grey correlation coefficient values corresponding to the peak moments for the fully irrigated, mild water stress, moderate water stress, and severe water stress treatments did not differ significantly. This indicated that the linear correlation [33], nonlinear correlation [34], and curve similarity [35] of Tc and Ta under the four moisture treatments after time-lag correction did not differ much."
Q5:All symbols and abbreviations need to be defined in the legend of the figures and tables.
Response:Thanks for your valuable suggestion.(1)We have replaced Figure 2 with the figure below. At the same time, we have replaced the caption of Figure 2 with "Canopy temperature of winter wheat under four moisture treatments."
(2)We have replaced the caption of Figure 4 with "Pn (µmol/(m²·s)), gs (mol/(m²·s)) and Tr (mmol/(m²·s)) in winter wheat."
(3)We have replaced the caption of Figure 5 with "Result of the linear fit of the VPD (kPa) and the CTD (℃) from 13:00 to 15:00."
(4)We have replaced the caption of Figure 8 with "Time lag parameters and corresponding coefficients for the fully irrigated treatment calculated by the time-lag cross-correlation method, time-lag mutual information method, and grey time-lag correlation analysis."
(5)We have replaced the caption of Figure 9 with "Time lag parameters and corresponding coefficients for the mild water stress treatment calculated by the time-lag cross-correlation method, time-lag mutual information method, and grey time-lag correlation analysis."
(6)We have replaced the caption of Figure 10 with "Time lag parameters and corresponding coefficients for the moderate water stress treatment calculated by the time-lag cross-correlation method, time-lag mutual information method, and grey time-lag correlation analysis."
(7)We have replaced the caption of Figure 11 with "Time lag parameters and corresponding coefficients for the severe water stress treatment calculated by the time-lag cross-correlation method, time-lag mutual information method, and grey time-lag correlation analysis."
(8)We have replaced the caption of Figure 12 with "Heat map of CWSI theoretical model and Pn before and after considering time lag effects."
(9)We have replaced the caption of Figure 13 with "Heat map of CWSI empirical model and Pn before and after considering time lag effects."
(10)We have replaced the caption of Figure 14 with "Heat map of CWSI theoretical model and Tr before and after considering time lag effects."
(11)We have replaced the caption of Figure 15 with "Heat map of CWSI empirical model and Tr before and after considering time lag effects."
(12)We have replaced the caption of Figure 16 with "Heat map of CWSI theoretical model and gs before and after considering time lag effects."
(13)We have replaced the caption of Figure 17 with "Heat map of CWSI empirical model and gs before and after considering time lag effects."
Conclusion
Q6:It needs to be figured out. Please use the present tense and state the most important outcome of your work.
Response: Thanks for your valuable suggestion.The conclusion section has been condensed and revised to the present simple tense. The modified version is as follows:"In this study, we investigate the impact of the time lag effect between Tc and Ta on the correlation between CWSI and photosynthetic parameters. The main conclusions are: (1) The magnitude of the time lag parameter between Tc and Ta in winter wheat, calculated by the four methods for the entire reproductive period, follows the order: grey time lag correlation analysis > time lag peak-seeking method > time lag mutual information method > time-lag cross-correlation method. All time lag parameters of the the severe water stress treatment experience a sudden decrease; (2) The CWSI empirical model is more sensitive to the time lag effect than the theoretical model. Time-lag correction, particularly using the time lag mutual information method, significantly improves the correlation between CWSI and photosynthetic parameters; (3) The GA-SVM machine learning algorithm provides the highest prediction accuracy for daily changes in the CWSI empirical model corrected with the time-lag mutual information method (R² = 0.982, RMSE = 0.017)."
Besides, we are adding references to add content.

Reviewer 2 Report
Comments and Suggestions for Authors
The manuscript plants-3023810 entitled "Influence of time lag effect between winter wheat canopy temperature and atmospheric temperature on the accuracy of CWSI inversion of photosynthetic parameters" submitted by Wang et al. describe and discuss an interesting experiment were different approches were tested to improve the prediction of CWSI in wheat.
Considering the topic and the emerging difficulties related to climate change and water scarcity this study can provide useful information for the future of agriculture.
In general, the experiment was carried out in approchiate way and well presented in the M&M section. Manuscript is of good quality with an introduction that well presents the state of the art, a clear results and their good discussion. Then, conclusionions are based on the achieved results.
In my opion the issues of this manuscript are:
1) revise keywords: avoid to repeat words already mentioned in the title.
2) In the M&M section the machine learning approch was not describe. The authors should implement a complete description.
3) The figure captions are poor and they must to be implemented with a complete and exhaustive description for each figure.
Comments on the Quality of English LanguageThe language quality is good and only minor revision on the style are needed.
Author Response
Dear editor and reviewer,
Thank you for offering us an opportunity to improve the quality of our submitted manuscript (Influence of time lag effects between winter wheat canopy temperature and atmospheric temperature on the accuracy of CWSI inversion of photosynthetic parameters, manuscript mumber: 3023810). The time and efforts you spent to suggest improvements to our paper are much appreciated. Thank you for your careful review, We have incorporated your comments and prepared the revision for your review.The comments of the editor and reviewers are addressed on an individual basis. We hope that the responses would meet with your approval. Changes made in revised manuscript are highlighted with yellow colour. Please find our responses to each of the comments point by point below.
Responses to Reviewer #2 (Manuscript ID: Plants- 3023810)
Q1:revise keywords: avoid to repeat words already mentioned in the title.
Response:Thanks for your valuable suggestion.We have replaced the keywords with time lag effects; winter wheat; CWSI; photosynthetic rate; transpiration rate; stomatal conductance.
Q2: In the M&M section the machine learning approch was not describe. The authors should implement a complete description.
Response:Thanks for your valuable suggestion. In the M&M section (2.4.10 Machine Learning Algorithms), we have added the following to the machine learning approach:
In this study, various machine learning and deep learning methods were employed to process and predict the Crop Water Stress Index (CWSI), including Genetic Algorithm Optimised Support Vector Machines (GA-SVM), Bayesian Optimised Long Short-Term Memory Neural Networks (Bayes-LSTM), Particle Swarm Algorithm Optimised Long Short-Term Memory (PSO-LSTM), Convolutional Bi-directional Long Short-Term Memory Neural Networks (CNN-BILSTM), Attention Mechanism Long Short-Term Memory Neural Networks (Attention-LSTM), and Attention Mechanism Gated Recurrent Units (Attention-GRU). GA-SVM optimizes SVM parameters using a genetic algorithm, effectively enhancing the model's classification and prediction performance, making it suitable for small but complex datasets. PSO-LSTM employs particle swarm optimization to find the optimal parameters for LSTM, improving prediction performance and training efficiency, suitable for scenarios with a large parameter space. CNN-BILSTM combines CNN and bi-directional LSTM to simultaneously extract spatial and temporal features, enhancing the prediction capability for complex long time series data with spatial dependencies. Attention-LSTM incorporates an attention mechanism into LSTM, enhancing the model's focus on important time steps and improving prediction accuracy, particularly for long time series data with significant features. Attention-GRU introduces an attention mechanism into GRU, simplifying the network structure while improving the focus on important time steps, making it suitable for efficient prediction of long time series data. Overall, the introduction of the attention mechanism (Attention-LSTM and Attention-GRU) significantly enhances the model's ability to capture important information, thus improving prediction accuracy. Bayes-LSTM enhances model robustness by addressing parameter uncertainty. Both PSO-LSTM and GA-SVM improve model performance through optimization algorithms but are sensitive to initial settings and optimization processes.
Q3: The figure captions are poor and they must to be implemented with a complete and exhaustive description for each figure.
Response:Thanks for your valuable suggestion.
(1)We have replaced Figure 2 with the figure below. At the same time, we have replaced the caption of Figure 2 with "Canopy temperature of winter wheat under four moisture treatments."
(2)We have replaced the caption of Figure 4 with "Pn (µmol/(m²·s)), gs (mol/(m²·s)) and Tr (mmol/(m²·s)) in winter wheat."
(3)We have replaced the caption of Figure 5 with "Result of the linear fit of the VPD (kPa) and the CTD (℃) from 13:00 to 15:00."
(4)We have replaced the caption of Figure 8 with "Time lag parameters and corresponding coefficients for the fully irrigated treatment calculated by the time-lag cross-correlation method, time-lag mutual information method, and grey time-lag correlation analysis."
(5)We have replaced the caption of Figure 9 with "Time lag parameters and corresponding coefficients for the mild water stress treatment calculated by the time-lag cross-correlation method, time-lag mutual information method, and grey time-lag correlation analysis."
(6)We have replaced the caption of Figure 10 with "Time lag parameters and corresponding coefficients for the moderate water stress treatment calculated by the time-lag cross-correlation method, time-lag mutual information method, and grey time-lag correlation analysis."
(7)We have replaced the caption of Figure 11 with "Time lag parameters and corresponding coefficients for the severe water stress treatment calculated by the time-lag cross-correlation method, time-lag mutual information method, and grey time-lag correlation analysis."
(8)We have replaced the caption of Figure 12 with "Heat map of CWSI theoretical model and Pn before and after considering time lag effects."
(9)We have replaced the caption of Figure 13 with "Heat map of CWSI empirical model and Pn before and after considering time lag effects."
(10)We have replaced the caption of Figure 14 with "Heat map of CWSI theoretical model and Tr before and after considering time lag effects."
(11)We have replaced the caption of Figure 15 with "Heat map of CWSI empirical model and Tr before and after considering time lag effects."
(12)We have replaced the caption of Figure 16 with "Heat map of CWSI theoretical model and gs before and after considering time lag effects."
(13)We have replaced the caption of Figure 17 with "Heat map of CWSI empirical model and gs before and after considering time lag effects."

Round 2
Reviewer 2 Report
Comments and Suggestions for Authors
The authors applied thre requeste revision to the manuscript improving its quality.
Comments on the Quality of English Languagethe language quality is good enough